# Synthesis of (*+*)*-*(*R*)-Tiruchanduramine

**DOI:** 10.3390/molecules27041338

**Published:** 2022-02-16

**Authors:** Zahraa S. Al-Taie, Barbara Bartholomew, Christopher Cartmell, Richard T. Froom, Russell G. Kerr, Rolf Kraehenbuehl, Patrick J. Murphy, Robert J. Nash, Yana B. Penkova, Alexander van Teijlingen

**Affiliations:** 1Department of Chemistry, College of Science, Al-Nahrain University, Baghdad 10072, Iraq; zahraasabah80@hotmail.com; 2The Institute of Biological, Environmental and Rural Sciences (IBERS), Aberystwyth University, Aberystwyth SY23 3DA, UK; barbara.bartholomew@phytoquest.co.uk (B.B.); robert.nash@phytoquest.co.uk (R.J.N.); y.penkova@phytoquest.co.uk (Y.B.P.); 3Department of Chemistry and Biomedical Sciences, DRC, 550 University Avenue, University of Prince Edward Island, Charlottetown, PE C1A 4P3, Canada; ccartmell@upei.ca (C.C.); rkerr@upei.ca (R.G.K.); 4School of Natural Sciences (Chemistry), Bangor University, Bangor LL57 2UW, UK; chpa30@bangor.ac.uk (R.T.F.); alexander.van-teijlingen@strath.ac.uk (A.v.T.); 5Centre for Environmental Biotechnology, Bangor University, Deiniol Rd., Bangor LL57 2UW, UK; r.kraehenbuehl@bangor.ac.uk

**Keywords:** *Synoicum macroglossum*, tiruchanduramine, β-carboline, guanidines

## Abstract

The absolute stereochemistry of the marine alkaloid (*+*)-(*R*)-tiruchanduramine was established via a convergent total synthesis in six steps and 15.5% overall yield from Fmoc-D-Dab(Boc)-OH.

## 1. Introduction

We have previously reported [1] a racemic synthesis of the alkaloid tiruchanduramine **1** (Figure 1) isolated by Ravinder et al. [2] from the ascidian *Synoicum macroglossum* and shown to be a α-glucosidase inhibitor. Glucosidases have a number of roles in the body, including digestion, glycoprotein processing and degradation, and have differing locations and substrate specificities [3]. They are targets for treatments to viral infections, diabetes, lysosomal storage disorders, and cancer [4]. Glucosidases involved in digestion have attracted particular interest as they release glucose and inhibiting them slows rises in blood sugar following food consumption and medications such as Acarbose [5] and Miglitol [6] have been developed.

We wished to determine the absolute configuration of **1** by synthesis from a chiral pool precursor and were initially attracted by the synthesis reported by Ravinder et al. [2]. This work reported the coupling of *β*-carboline-3-carboxylic acid **2** [7], easily prepared in four steps from tryptophan, with the amine **3**, prepared in five steps from but-3-en-1-ol. Amide **4** was converted into the hydrochloride salt of **1** via a four-step protocol involving the introduction of the guanidine under Mitsunobu conditions (Figure 1). Our racemic synthesis [1] differed from this as it was a convergent approach in which the guanidine heterocycle **5** is prepared separately then coupled to the *β*-carboline-3-carboxylic acid **2** (Figure 1).

## 2. Results and Discussion

We thus attempted to repeat the work reported by Ravinder [2] by preparing the known [8] chiral amine (*S*)-**3** from commercially available (*S*)-malic acid **6** using literature procedures (esterification [8] (MeOH, H^+^), reduction [8] (H_3_B·SMe_2_, NaBH_4_, THF), protection [8] (2-methoxypropene, H^+^), reduction [8] (LiAlH_4_, Et_2_O), azide formation [9] (TPP, DIAD, DPPA), and reduction [10] (TPP, H_2_O), see Appendix A for experimental details). Coupling of the amine (*S*)-**3** with carboxylic acid **2** was attempted using DCC and DMAP as reported by Ravinder [2]. Whilst successful, the reaction was hampered by difficulties associated with the removal of the DCCU byproduct from the desired product **7**. To circumvent this, we repeated the reaction by activating the acid **2** with CDI followed by the addition of the amine (*S*)-**3**. After 24 h, work up and chromatography gave **7** in a 70% yield and the structure was confirmed by NMR and MS analysis. Following this, the *β*-carboline NH was protected using Boc-anhydride and trimethylamine, to secure **8** in a 56% yield, again, its structure was confirmed by NMR and MS analysis. Finally, deprotection of the acetonide using PPTSA in methanol gave **9** in 71% yield. NMR data had been reported [2] for this compound and comparison of this gave a close agreement despite several signals being missing in the original literature report. MS data was as expected for the desired compound. With diol **9** in hand, we attempted the formation of the cyclic guanidine **10** using the conditions suggested in the original racemic synthesis. [2] Unfortunately, in our hands we were unable to effect any reaction between **9** and N,N′,N′′-tri-Boc-guanidine **11** using DEAD and triphenylphosphine. We varied the concentration of the reaction and the number of equivalents of each reagent and used the alternate reagent DIAD as well as differing solvents. In all cases only the unreacted diol **9** was recovered after chromatography (Figure 2).

Frustrated by this synthetic approach, we returned to our original convergent strategy and sought a synthetic route to enantiopure guanidine **5**. Our synthesis began with the commercially available orthogonally protected diaminocarboxylic acid Fmoc-D-Dab(Boc)-OH (**12**). Reduction of the mixed anhydride of **12** with sodium borohydride gave alcohol **13** in an 86% yield [11]. Deprotection of **13** using piperidine in dichloromethane gave the amine **14** which was guanidinylated [1] with **15** to give the alcohol **16** in 94% yield. Cyclization of this with a mixture of I_2_/dppe/PPh_3_/imidazole [1] in dichloromethane to give the cyclic guanidine **17** which was used in the next steps without purification as it was found to undergo decomposition on silica gel. Crude **17** was deprotected using aqueous HCl (3M) and after freeze drying gave the desired guanidine (*R*)-**5**. This compound gave identical data to the previously reported racemic compound [1] and the synthetic methods adopted have preserved the original stereogenic center found in compound **12**. The preparation of (*R*)-**1** was achieved by treatment of the carboxylic acid **2** with an excess of 1,1′-carbonyldiimidazole (CDI) in DMF for 30 min followed addition of the free base of (*S*)-**5**. [1] After 24 h the mixture was freeze dried and purified by repeated column chromatography to give (*R*)-**1**·HCl in 18% yield over three steps. Synthetic **1**·HCl gave identical NMR data to that previously reported [1] for racemic **1**·HCl, however it gave an [α]_D_^24^ of +27 (MeOH, *c* = 1.0,) which is in close agreement to the value for natural **1** which was reported [2] as [α]_D_ +31 (c 0.5, MeOH) with an unspecified counterion (Figure 3). 

Ravinder^2^ reported that tiruchanduramine **1** was a potent inhibitor of yeast α-glucosidase (IC_50_ 78.2 μg/mL, 0.24 mM) as compared with Acarbose (IC_50_ 100 μg/mL, 0.16 mM) as the standard. We have previously reported that racemic tiruchanduramine **1** is a weak inhibitor of yeast α-glucosidase and now with a full synthesis of the natural product (*+*)-(*R*)-tiruchanduramine **1**, we confirm that the compound is not a potent inhibitor of either yeast α-(IC_50_ 0.7 mM) or almond β-glucosidase (IC_50_ 1.0 mM). Our assays show the same order of magnitude inhibition to the reported figures which correlates with Ravinder’s report, but not with the conclusion that the compound is a potent inhibitor for which much lower µM IC_50_ values would be expected. If there is a greater inhibition seen by Ravinder then we propose that, as with many natural products, the original natural product may have had a minor contaminant which produced the greater apparent inhibition. The comparison with acarbose by Ravinder may not have been appropriate since Acarbose does not potently inhibit yeast α-glucosidase [12]. Ravinder does not report which specific α-glucosidase they investigated but cite the method of Kim et al. who used a yeast α-glucosidase [13]. It is noteworthy that the synthetic analogues of tiruchanduramine **1** we prepared [14] gave considerably better IC_50_ values against yeast α-glucosidase, for example compounds **18a** (*n* = 4) and **18b** (*n* = 5) IC_50_ values of 12 µM and 7.5 µM respectively. (Figure 2) We can thus propose that although the natural product is not a very good inhibitor of yeast α-glucosidase, it represents a useful starting point for the design of much more potent inhibitors. 

## 3. Conclusions

We have succeeded in the first enantioselective synthesis of (+)-(*R*)-tiruchanduramine **1** as its hydrochloride and formate salts in 15.5% overall yield (6 steps) from a commercially available starting material **12** and have established its absolute stereochemistry as *R*. We also attempted to repeat reactions reported in the original synthesis which proved capricious in our hands. 

## 4. Experimental

Column chromatography was carried out on silica gel (60Å, 40–63 μm) and TLCs were conducted on precoated Kieselgel 60 F254 (Art. 5554) (Merck Life Science UK Limited, Watford, UK) with the eluent specified in each case. All non-aqueous reactions were conducted in oven-dried apparatus under a static atmosphere of argon. Diethyl ether, THF, and dichloromethane were dried by a Pure Solv MD-3 solvent purification system (One Industrial Way, Amesbury, MA, USA). Dry methanol and DMF and all starting materials were purchased from Aldrich (Merck Life Science UK Limited, Gillingham, UK). Chemical shifts are reported in δ values relative to residual chloroform (7.26/77.16 ppm), methanol (3.31/49.0 ppm), and DMSO (2.50/39.52 ppm) as internal standards. Proton and carbon NMR spectra were recorded in CDCl_3_ on a Bruker AC400 spectrometer (Coventry, UK) unless otherwise stated and spectra are presented in the Appendix A for each compound. Mass spectra data were obtained at the EPSRC Mass Spectrometry Service Centre at the University of Wales, Swansea. Infrared spectra were recorded as thin films (oils) on a Bruker Tensor 27 series instrument (Coventry, UK). Optical rotations were performed on a Bellingham and Stanley Ltd. ADP400^TM^ polarimeter (Bellingham+Stanley Ltd, Kent, UK). Melting points were performed on a Stuart SMP10 apparatus (Cambridgeshire, UK) and are uncorrected.

### 4.1. S-N-(2-(2,2-Dimethyl-1,3-dioxolan-4-yl)ethyl)-9H-pyrido [3,4-b]indole-3-carboxamide **7**


Carboxylic acid **2** (209 mg, 0.98 mmol, 1.0 equiv.) and CDI (340 mg, 2.10 mmol, 2.13 equiv.) were dissolved in dry DMF (5 mL) and the solution stirred for 30 min at which point the acid **2** had dissolved. Amine (*S*)-**3** (1.00 g, 6.89 mmol. 7.0 equiv.) was then added as a solution in DMF (2 mL) and this mixture was stirred for 24 h. Water (10 mL) was then added and stirring continued for 1 h. The mixture was extracted with DCM (3 × 50 mL) and the combined extracts washed with water (3 × 100 mL). After drying (MgSO_4_) and evaporation, chromatography on silica gel (50% EA in CF) gave amide **7** (234 mg, 0.69 mmol, 70%) as an amorphous white solid. [α]_D_^21^ −13.3 (c = 0.42, CH_2_Cl_2_); Mp 194–195 °C; Rf 0.14 (50% EA in CF); **δ_H_** 9.50 (1H, s, NH), 8.91 (1H, s, CH), 8.80 (1H, s, CH), 8.51 (1H, t, *J* 5.9 Hz, CH), 8.13 (1H, d, *J* 7.9 Hz, CH), 7.55–7.56 (2H, m, 2 × CH), 7.31 (1H, dt, *J* 8.0, 4.0 Hz, CH), 4.23–4.29 (1H, m, CH), 4.10 (1H, dd, *J* 8.0, 6.0 Hz, 1H), 3.59–3.75 (3H, m, CH_2_, CH), 2.01–1.87 (2H, m, CH_2_), 1.45 (3H, s, Me), 1.37 (3H, s, Me); **δ_C_** 165.9, 141.0, 140.2, 137.4, 131.9, 129.7, 129.0, 122.2, 121.9, 120.9, 114.5, 112.0, 109.2, 74.6, 69.4, 36.9, 33.7, 27.1, 25.8; ***ν*_max_** (film)/cm^−1^ 3386, 3160, 3032, 2983, 2877, 1651, 1624, 1592, 1533, 1499, 1481, 1461, 1406, 1368, 1338, 1311, 1255, 1209, 1161, 1112, 1103, 1045, 1016, 986; MS (ESI) *m*/*z* 703.3 (65% [2M + Na]^+^), 362.1 (100% [M + Na]^+^), 240.2 (35% [M + H]^+^); HMRS (ESI) found 340.1661, C_19_H_22_N_3_O_3_^+^ ([M + H]^+^) requires 340.1656.

### 4.2. Tert-Butyl S-3-((2-(2,2-dimethyl-1,3-dioxolan-4-yl)ethyl)carbamoyl)-9H-pyrido[3,4-b]indole-9-carboxylate **8**

Acetonide **7** (139 mg, 0.41 mmol, 1.0 equiv.) was dissolved in dry DCM (10 mL), triethylamine (0.12 mL, 83 mg, 0.82 mmol, 2.0 equiv.) and Boc anhydride (179 mg, 0.82 mmol, 2.0 equiv.) were added and the mixture stirred overnight. Evaporation under reduced pressure followed by column chromatography (30–50% EA in PE) gave **8** (101 mg, 0.23 mmol) in 56% yield as a pale-yellow solid. Rf 0.17 (40% EA in PE); [α]_D_^21^ −21.0 (*c* = 0.19, CH_3_OH); Mp.133–136 C; **δ_H_** 9.48 (1H, d, *J* 0.7 Hz, CH), 8.81 (1H, d, *J* 0.7 Hz, CH), 8.51 (1H, br t, *J* 5.6 Hz, NH), 8.37 (1H, d, *J* 8.5 Hz, CH), 8.14 (1H, d, *J* 7.6 Hz, CH), 7.64 (1H, ddd, *J* 1.0, 7.6, 8.5 Hz, CH), 7.46 (1H, t, *J* 7.6 Hz, 1H), 4.25–4.32 (1H, m, CH), 4.12 (1H, dd, *J* 6.1, 8.0 Hz, CH), 3.60–3.75 (2H, m, CH_2_), 3.64 (1H, dd, *J* 7.1, 8.0 Hz, CH), 1.87–2.03 (2H, m, CH_2_), 1.79 (9H, s, 3 × Me), 1.47 (3H, s, Me), 1.39 (3H, s, Me); **δ_C_** 164.8, 150.3, 143.7, 139,7, 136.4, 133.1, 130.3, 124.1, 121.7, 116.7, 113.6, 109.2, 85.5, 74.7, 69.4, 37.0, 33.6, 28.5, 27.1, 25.8; ν_max_ 3323, 2981, 2932, 2867, 1728, 1651, 1620, 1563; MS (ESI) *m*/*z* 440.2 (100%, [M + H]^+^); HMRS (ESI) found 440.2183, C_24_H_30_N_3_O_5_^+^ ([M + H]^+^) requires 440.2180.

### 4.3. Tert-Butyl (S)-3-((3,4-dihydroxybutyl)carbamoyl)-9H-pyrido[3,4-b]indole-9-carboxylate **9**

Acetonide **8** (100 mg, 0.23 mmol, 1.0 equiv.) was dissolved in MeOH (2.5 mL), pyridinium *p*-toluenesulfonate (23 mg, 0.09 mmol 0.4 equiv.) was added and the mixture stirred for 24 h. After evaporation under reduced pressure, chromatography (25–100% EA in PE) gave the diol **9** (65 mg, 0.16 mmol) in 71% yield as a pale-yellow solid. [α]_D_^16^ +7.2 (c = 0.5, CHCl_3_); Mp 123–125 °C; Rf 0.46 (10% ME in EA); **δ_H_** 9.48 (1H, s, CH), 8.80 (1H, s, CH) 8.51 (1H, br t, *J* 6.5 Hz, NH, 8.36 (1H, d, *J* 8.5 Hz, CH), 8.12 (1H, d, *J* 7.7 Hz, CH), 7.65 (1H, ddd, *J* 1.2, 7.3, 8.5 Hz, CH), 7.46 (1H, dt, *J* 1.2,7.3 Hz, 1H), 3.99–4.08, (1H, m, CH), 3.77–3.83 (1H, m, CH), 3.65 (1H, dd, *J* 11.2, 3.2 Hz, 1H), 3.54 (1H, dd, *J* 11.2, 7.5 Hz, 1H), 3.41–3.49 (1H, m, CH), 1.79 (11H, s, 3 × Me, 2 × OH), 1.67–1.74 (2H, m, CH_2_); **δ_C_** 166.3, 150.2, 143.0, 139.6, 136.6, 136.4, 133.0, 130.3, 124.1, 123.8, 121.6, 116.7, 113.8, 85.5, 69.1, 66.7, 36.1, 33.6, 28.4; **ν_max_** 3413, 3302, 2942, 1732, 1666, 1619, 1562, 1527, 1491, 1454, 1414, 1395, 1370, 1354, 1325, 1276, 1248, 1219, 1202, 1152, 1119, 1082, 1064, 1032, 1018, 986; MS (ESI) *m*/*z* 400.2 (100% [M + H]^+^); HMRS (ESI) found 400.1865, C_21_H_26_N_3_O_5_^+^ ([M + H]^+^) requires 400.1867.

### 4.4. (9. H-Fluoren-9-yl)methyl tert-butyl (4-hydroxybutane-1,3-diyl)-(R)-dicarbamate **13**

Ethyl chloroformate (0.50 mL, 497 mg, 5.22 mmol, 1.1 equiv.) was added to a cooled (−5° C) and stirred solution of Fmoc-D-Dab(Boc)-OH **12** (2.09 g, 4.74 mmol, 1.0 equiv.) and triethylamine (0.73 mL, 730 mg, 5.22 mmol, 1.0 equiv.) in dry THF (20 mL). The reaction was stirred at −10 to −5 °C for 30 min and the precipitated triethylamine hydrochloride was removed by filtration and the filter pad washed with dry THF (10 mL). The combined filtrates were cooled (−5 °C) and added in a dropwise fashion to a cooled (−5 °C) solution of sodium borohydride (538 mg, 14.2 mmol, 3.0 equiv.) in a mixture of water and THF (10 mL, 1:1). The reaction mixture was stirred at this temperature for 2.5 h and then allowed to warm to rt overnight. The solvent was evaporated under vacuum, the residue diluted with water (100 mL) then extracted with ethyl acetate (3 × 100 mL). The combined organic extracts were washed with citric acid (100 mL, aq., 10% *w*/*v*), NaHCO_3_ (100 mL, aq. 5% *w*/*v*), brine (100 mL, aq., sat.), water (100 mL) and then dried (MgSO_4_). After evaporation under reduced pressure, purification by column chromatography (gradient elution, 0–90% EA in PE) gave **13** (1.74 g, 4.08 mmol) as a white solid in 86% yield [11]. Rf 0.40 (EA); **[α]_D_^22^** +36.0 (CH_3_Cl, c 1.6); Mp 99–101 °C; **δ_H_** δ 7.76 (2H, d, *J* 7.5 Hz, 2 × CH), 7.60 (2H, d, *J* 7.3 Hz, 2 × CH), 7.40 (2H, app. t, *J* 7.4 Hz, 2 × CH), 7.31 (2H, app. t, *J* 7.4 Hz, 2 × CH), 5.61 (1H, br s, NH), 5.26 (1H, br s, NH), 4.46 (1H, dd, *J* 6.9, 10.0 Hz, CH), 4.38 (1H, dd, *J* 3.4, 10.0 Hz, CH), 4.15–26 (1H, m, CH), 3.55–3.80 (3H, m, CH and CH_2_), 3.18–3.52 (2H, m, CH_2_), 2.85–3.00 (1H, m, CH), 1.55–1.79 (2H, m, CH_2_), 1.46 (s, 9H) ppm; **δ_C_** 157.0, 156.4, 143.9, 141.4, 127.8, 127.1, 125.1, 120.0, 79.5, 66.6, 64.8, 50.5, 47.3, 37.2, 32.0, 28.5 ppm; **ν_max_** 3346, 3031, 2973, 1682, 1524, 1477, 1448, 1167 cm^−1^; MS (ESI) *m*/*z* 427.2 (40%, [M + H]^+^), 327.2 (100%, [M + H-Boc]^+^); HRMS (ESI) m/z C_24_H_31_N_2_O_5_ [M + H]^+^), requires 427.2227, found 427.2227.

### 4.5. Tert-Butyl (R)-(4-hydroxy-3-(bis-tert-butyl-guanidine)butyl)carbamate **16**

Alcohol **13** (1.74 g, 4.08 mmol, 1.0 equiv.) was dissolved in 20% piperidine in chloroform (*v*/*v*, 40 mL) and stirred. After 2 h TLC (100% EA) indicated the reaction was complete and the solvent was evaporated under high vacuum. The resulting white solid was triturated with hexane (2 × 50 mL) to remove Fmoc by-products. After drying, the resulting crude amine **14** and compound **15** (1.90 g, 6.12 mmol, 1.5 equiv.) were dissolved in dry methanol (20 mL) and NEt_3_ (1.7 mL, 1.24 g, 12.2 mmol, 3.0 equiv.) was added. After stirring for 48 h, the solvent was removed under reduced pressure and the crude product purified by column chromatography (30–40% EA in PE) to give **16** (1.72 g, 3.85 mmol) as a white solid (recrystallised from Et_2_O/hexane) in 94% yield. Mp 148 °C; Rf 0.13 (30% EA in PE); **[α]_D_^23^** +72 (CH_3_Cl, *c* = 1.2); **δ_H_** 11.35 (1H, s, NH), 8.59 (1H, d, *J* 7.7 Hz, NH), 5.83–5.92 (1H, m, NH), 4.11–4.24 (1H, m, CH), 3.97 (1H, br s, OH) 3.68 (1H, dd, *J* 11.0, 3.3 Hz, CH), 3.58 (1H, dd, *J* 11.0, 4.5 Hz, CH), 3.32–3.46 (1H, m, CH), 2.75–2.91 (1H, m, CH), 1.56–1.76 (2H, m, CH_2_), 1.44 (9H, s, CH_3_), 1.42 (9H, s, CH_3_), 1.38 (9H, s, CH_3_); **δ_C_** 162.7, 156.6, 156.4, 152.9, 83.5, 79.5, 78.9, 65.2, 50.1, 36.9, 32.4, 28.5, 28.3, 28.1; **ν_max_** 3297, 3273, 3969, 2906, 1718, 1655, 1607, 1543, 1413, 1392, 1367, 1353, 1324, 1305, 1273, 1249, 1235, 1150, 1121, 1072, 1048, 1039, 806, 778, 700; MS (ESI) *m*/*z* 447.3 (100%, [M + H]^+^); HRMS (ESI) C_20_H_39_N_4_O_7_ requires 447.2813 found 447.2812.

### 4.6. Tert-Butyl (R)-4-(2-((tert-butoxycarbonyl)amino)ethyl)-2-((tertbutoxycarbonyl)imino)imidazolidine-1-carboxylate **17**

Guanidine **16** (740 mg, 1.66 mmol, 1.0 equiv.) was dissolved in dichloromethane (15 mL) and cooled (−40 °C) whereupon dppe (990 mg, 2.49 mmol, 1.5 equiv.) was added in one portion and the mixture stirred until complete dissolution was achieved. Imidazole (395 mg, 5.80 mmol, 3.5 equiv.) and iodine (631 mg, 2.50 mmol, 1.5 equiv.) were then added sequentially and the resultant mixture stirred for 3 h with cooling (−40 to −20 °C). Chloroform (50 mL) was then added, followed by ammonium chloride solution (aq., sat., 100 mL). After warming to rt, the organic layer was separated, and the aqueous layer extracted with chloroform (2 × 20 mL). The combined extracts were washed with water (50 mL) and brine (50 mL) then dried (MgSO_4_), filtered and concentrated under reduced pressure. The resulting residue was dissolved in a minimum volume of dichloromethane (4–5 mL) and then diethyl ether was added to precipitate any DPPE by-products. The supernatant liquid was removed by filtration and after evaporation this process of precipitation was repeated twice. This gave **17** (555 mg, contaminated with dppe byproducts) as a sticky white gum which was used in the next step without further purification. **[α]_D_^24^** + 22.5 (CHCl_3_, c = 1.0); **δ_H_** 5.25–5.21 (1H, br s, NH) 4.02–4.18 (1H, m, CH), 3.85–3.96 (1H, m, CH), 3.17–3.45 (3H, m, CH, CH_2_), 1.70–1.87 (2H, m, CH_2_), 1.51 (9H, s, CH_3_), 1.49 (9H, s, CH_3_), 1.43 (9H, s, CH_3_) guanidine NH not detected; **δ_C_** 156.4, 154.5, 152.9, 151.3, 83.9, 81.7, 79.2, 50.4, 48.4, 37.7, 36.5, 28.5, 28.2, 28.2; **ν_max_** 3328, 2906, 2923, 2844, 1697, 1511, 1391, 1366, 1304, 1248, 1140, 1042, 1007, 773, 729; MS (ESI) *m*/*z* 429.3 (100%, [M + H]^+^); HRMS (ESI) C_20_H_37_N_4_O_6_ ([M + H]^+^) requires 429.2708 found 429.2707. 

### 4.7. (R)-2-(2-Iminoimidazolidin-4-yl)ethan-1-amine Dihydrochloride

Hydrochloric acid solution (5.0 mL, aq, 3M) was added to a cooled (0 °C) flask containing guanidine **17** (488.0 mg) and the mixture stirred to rt over 48 h. The mixture was passed through a short Celite© pad in a pipette to remove dppe derived impurities and the pad washed with water (ca. 2–3 mL). The filtrate was freeze dried to give crude (*R*)-**5** (259 mg) as a hygroscopic off-white solid which was used in next step without any further purification. **[α]_D_^24^** + 24 (H_2_O, *c* = 2.0); **δ_H_** (D_2_O) 4.15–4.22 (1H, m, CH), 3.86 (1H, app t, *J* 9.9 Hz, CH), 3.43 (1H, dd, 6.4, 9.9 Hz, CH), 3.02–3.15 (2H, m, CH_2_), 1.99–2.05 (1H, m, CH_2_); **δ_C_** (D_2_O) 159.4, 52.6, 47.6, 35.7, 31.8; MS (ESI) *m*/*z* 129.1 (100%, [M + H]^+^); HRMS (ESI) C_5_H_13_N_4_ requires 129.1135, found 129.1135.

### 4.8. (+)-(R)-Tiruchanduramine **1**

Carboxylic acid **2** (257 mg, 1.22 mmol, 1.03 equiv.) was suspended in anhydrous DMF (7 mL) and CDI (258 mg, 1.72 mmol, 1.45 equiv.) was added and the mixture stirred for 30 min. This solution was then transferred via syringe to a cooled (0 °C) mixture of crude (*R*)-**5** (238 mg, 1.18 mmol) and triethylamine (0.35 g, 0.5 mL, 3.5 mmol, 3.0 equiv.) in DMF (8 mL). After stirring for 24 h, the mixture was freeze dried to give a crude product (0.75 g). This compound was purified by column chromatography (0–20% MeOH in CHCl_3_). The fractions containing **1** were combined and dissolved in MeOH (ca 5–10 mL) containing a drop of concentrated HCl and filtered through a plug of cotton wool. After evaporation the resulting solid (155 mg) was repeated purified by column chromatography (4–8% MeOH in CHCl_3_; top/tail method) to give (*R*)-**1**·HCl (86 mg, 0.24 mmol, 18% yield over three steps, ca. 90–95% pure) as an off white solid. Rf 0.11 (20% MeOH in CHCl_3_); **[α]_D_^24^** + 27 (MeOH, *c* = 1.0, lit.^2^ **[α]_D_** + 31 (MeOH, *c* = 0.5)); Mp 183–185 °C. A sample (20 mg, 8 × 2.5 mg injections) was purified by reverse-phase HPLC using a Luna 100A C18 column (5 µm 250 × 10 mm, Phenomenex). Over 50 min a gradient elution was performed with initial conditions of 98% Solvent A (MQ water, 0.1% formic acid) 2% Solvent B (MeOH 0.1% formic acid) which was held for 1 min. A gradient to 95% solvent B was performed over 35 min, which was held for 5 min before returning to initial conditions over the next 5 min which were then held for a further 5 min to equilibrate the column prior to the next injection. (+)-(*R*)-Tiruchanduramine **1**·HCO_2_H (10 mg) eluted at 27 min as a cream colored wax **[α]_D_^22^** + 27 (MeOH, *c* = 0.2, lit. [2] **[α]_D_** + 31 ((*c* = 0.5, MeOH)); Spectroscopic data was in agreement with the literature data [2]. 

Data for **1**·HCO_2_H: **δ_H_** (D_6_-DMSO) 1.82 (2H, apparent q, *J* 6.6 Hz, CH_2_), 3.25 (1H, dd, *J* 7.0, 9.4 Hz, CH), 3.35–3.51 (2H, m, CH_2_), 3.72 (1H, apparent t, *J* 9.4 Hz, CH), 3.92–3.99 (1H, m, CH), 7.30 (1H, t, *J* 7.4 Hz, CH), 7.60 (1H, t, *J* 7.6 Hz, CH), 7.66 (1H, d, *J* 8.2 Hz, CH), 8.33 (2H, br s, NH, CH), 8.37 (2H, br s, 2 × NH), 8.38 (1H, d, *J* 7.9 Hz, CH) 8.83 (1H, br s, CH), 8.90 (1H, t, *J* 6.2 Hz, NH), 8.89 (1H, br s, CH), 8.98 (1H, br t, *J* 6.2 Hz, NH), 12.05 (1H, br s, NH); **δ_C_** (D_6_-DMSO) 35.1, 35.4, 47.9, 53.0, 112.3, 114.0, 120.0, 121.0, 122.2, 128.2, 128.6, 132.3, 137.2, 139.6, 141.1, 159.9, 165.1 165.5 (formate CH); MS (ESI) *m*/*z* 323.2 (100,% [M + H]^+^) HRMS (ESI) found 323.1612, C_17_H_19_N_6_O ([M+H]^+^) requires 323.1615.

### 4.9. Method for Determining Glycosidase Activity

Yeast α-glucosidase and almond β-glucosidase and the corresponding p-nitrophenyl-substrates were purchased from Sigma and assays were conducted as described previously [14,15]. Compounds were dissolved in water for inclusion in the assays. 

## Data Availability

Not applicable.

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
