# Peer review of "Synthesis of (+)-(R)-Tiruchanduramine"

_molecules, 2022, doi:10.3390/molecules27041338_

Round 1

Reviewer 1 Report

The presented manuscript dedicate to the preparation of marine alkaloid (+)-(R)-tiruchanduramine. It is nice synthetic work with sound chemistry. All isolated compounds were adequately characterized by spectral methods. I think it is suitable for publication in Molecules after some revision:

  1. The Introduction must be improved by the presentation α-glucosidase inhibitors’ importance.
  2. The Results and Discussion part should be separated from the Introduction. In the present form, the manuscript is not clear enough.
  3. “ml” should be changes to “mL”: Page 4, lines 137 and 139; Page 5, line 186 and 193; Page 7, line 266;
  4. For some mass spectra the ionization method is mentioned, whereas for some others not. It should be mentioned for all MS.

Reviewer 2 Report

In this work, entitled Synthesis of (+)-(R)-Tiruchanduramine, the authors present a new synthetic approach to this alkaloid. The proposal is based on previous syntheses carried out in a new stereoselective version. The achievement of the target product has allowed the authors to determine the absolute configuration of the natural alkaloid.

I think this is a work that could finally be published in the Molecules journal after some significant amendments. First of all, in my opinion, it should be pointed along the text that the stereogenic center of the starting commercial amino acid, 12, does not undergo any racemization all through the synthesis. For this, when the specific rotation of the different intermediates is determined, when possible, a citation should be provided where it is described for those known intermediates(eg intermediate 13).

Also, the reader discovers within the Conclusions section that other analogs have been synthesized for which data on their inhibitory activity is briefly given. Given that this activity even exceeds that of (+)-(R)-Tiruchanduramine, I think it would be convenient to introduce it as part of the article (Results section) and explain it conveniently. In my opinion, this would considerably increase the quality of the work and the merit of its publication in this journal.

The supporting information presents spectra with very good quality. Also, the article is well written. One can follow it perfectly. Still, I think there is some fix that should be applied:

In the title, the parentheses are italicized, like the + and R descriptors, and they shouldn't be.

Contrary to the Introduction and the Experimental sections, there is no title for the Results and Discussion one.

Between lines 30 and 31 there is a period that must be a type error.

Line 68, after -OH 12 should be written (Scheme 3).

The specific rotation values, even if they are positive, should be expressed with their corresponding sign. This does not appear to be consistent throughout the work. Even in the Experimental section, only the value for diol 9 appears positive. This should be reviewed.

Line 113, there is a misplaced period.

Line 169 the reaction is actually started with acetonide 8, instead of 9. This needs to be changed.

Line 185, 12 should be marked in bold.

Line 311, a space is missing between 2005 and 46.

Line 319, a period is missing at the end of the reference.

All in all, In my opinion the work deserves to be published once the changes suggested above have been made.

Reviewer 3 Report

The presented chiral pool synthetic approach to natural product tiruchanduramine fulfilled its purpose. The absolute configuration of the natural product has been confirmed by the short total synthesis described in the article. Although no significant novelty (in terms of new methods) has been introduced the authors smartly utilized existing tools of organic synthesis. The work is very well presented and the article will be a joy to read for all organic chemists.

Author Response

Thanks for your positive comments on our article.  We had improved our paper.

Round 2

Reviewer 1 Report

In my opinion, "1.HCl" (Page 3, lines 83 and 84) is not a suitable form. It seems like the "space" is missed. Should be "1•HCl".

Reviewer 2 Report

Thanks for following my suggestions. I believe that now the work meets the required levels to be published in the journal Molecules.